# An Assessment of the Clinical Efficacy of a Topical Application of 5% Thymoquinone Gel for Plaque-Induced Gingivitis Patients: A Randomized Controlled Clinical Trial

**DOI:** 10.3390/healthcare12181898

**Published:** 2024-09-21

**Authors:** Ahmad H. Almehmadi, Khalid Aljohani

**Affiliations:** 1Department of Oral Biology, Faculty of Dentistry, King Abdulaziz University, Jeddah 21589, Saudi Arabia; 2Department of Oral Diagnostic Sciences, Faculty of Dentistry, King Abdulaziz University, Jeddah 21589, Saudi Arabia; koalgehani@kau.edu.sa

**Keywords:** thymoquinone, gingivitis, clinical trial

## Abstract

Background: Gingival diseases, encompassing a spectrum of oral health concerns, represent a prevalent issue within the global population. Despite their widespread occurrence, the research landscape concerning effective interventions, particularly those rooted in herbal products, remains somewhat limited. Addressing this knowledge gap, the current study undertook a comprehensive evaluation aimed at assessing the clinical efficacy of a novel intervention: a 5% thymoquinone (TQ) gel. This investigation specifically focused on the application of TQ gel as an adjunctive measure to the standard protocol of scaling (SC) in individuals afflicted with plaque-induced gingivitis. Through rigorous examination and analysis, this study seeks to provide valuable insights into the potential utility and therapeutic benefits of this herbal-based intervention in managing gingival diseases. Objective: To evaluate the efficacy of 5% TQ gel using a novel liposome drug delivery as a topical application following SC in gingivitis patients. Methods: A double-blinded, parallel, randomized controlled clinical trial. The study was performed at the Faculty of Dentistry, King Abdulaziz University, and Qassim University, Saudi Arabia. This trial enrolled 63 participants in an age group between 18 and 40 years attending the outpatient clinics of the Faculty of Dentistry, Qassim University, Saudi Arabia, and a clinical diagnosis of gingivitis was made. The enrolled subjects were categorized into three groups: Group I—TQ gel with SC, Group II—Placebo with SC, and Group III—SC alone, and clinical outcomes were measured at baseline and two-week follow-up visits. Plaque index (PI), papillary bleeding index (PBI), and any adverse events with TQ gel are categorized as mild, moderate, and severe. 63 patients. Group I (n = 21); Group II (n = 21); Group III (n = 21). Results: The paired *t*-test compared the mean differences in PI and PBI at two time points and it was observed that there were significant differences in Group I with *p*-values of 0.04 and 0.05, respectively. A one-way ANOVA test was performed and it showed significant differences in the mean scores between the three groups for PI (*p*-value—0.01) and PBI (*p*-value—0.05). The post hoc Tukey’s test compared the mean differences in PI and PBI between the groups and the results were in favor of Group I which used TQ gel with SC. Conclusions: The clinical trial concluded that the plaque and gingival bleeding scores were significantly reduced in the group of patients who intervened with TQ gel following SC when compared to SC-alone and placebo groups. Also, there were significant reductions in the scores from the baseline to the two-week follow-up visit in patients treated with TQ gel and SC.

## 1. Introduction

Gingivitis is the mildest form of gum disease, unlike periodontitis. It is often associated with irritation, swelling, and redness of the gum. However, periodontitis is a more serious form of gum disease that can damage the gums, bones, and ligaments that support the teeth. Gingivitis is considered the most widespread periodontal disease that affects the emerging dental layer [1,2,3]. It is shown to have a prevalence rate of 82% in the United States of America and nearly 100% in adult subjects aged 18 to 40 in Saudi Arabia [1]. Among the established causative factors of gingivitis and the subsequent progression to periodontal disease, dental plaque emerges as a well-documented culprit [4]. The pivotal trigger for initiating gingivitis, which can ultimately advance into periodontitis and precipitate tooth loss in the absence of intervention, remains the dental plaque matrix, which fosters diverse colonies of microorganisms [5].

While dental plaque forms the primary etiological foundation, various contributory elements have been identified in the onset of periodontal disease. The recent literature has unveiled the potential role of oxidative stress in the pathogenesis of periodontal disease [6,7,8,9,10]. Oxidative stress prominently contributes to the transition from gingival to chronic inflammatory periodontal disease, which is often provoked by bacterial stimuli from dental plaque [10,11,12]. Studies have probed end products of lipid peroxidation, including malondialdehyde and 8-isoprostane, in bodily fluids such as saliva, gingival crevicular fluid (GCF), and serum, revealing elevated levels in chronic periodontitis patients [13,14,15,16,17]. Additionally, heightened DNA damage marked by elevated 7–8-dihydro-8-oxoguanine (8-oxoG) levels—associated with ROS-mediated DNA degradation—has been discerned in gingival blood samples of chronic periodontitis patients [18]. Notably, serum reactive oxygen metabolite levels have displayed a significant correlation with antibody titers against *Porphyromonas gingivalis*, *Prevotella intermedia*, and *Eikenella corrodens* [19,20,21,22].

*Nigella sativa* (NS) and its constituent thymoquinone (TQ) display abundant potential therapeutic properties in diverse oral conditions [23]. Besides its anticariogenic effects [24], NS and TQ play a significant role in the prevention and treatment of periodontal diseases [25], as previewed in the current evaluation. Frequent examinations have evaluated the sensitivity of periodontal and oral pathogens against NS extracts and TQ and displayed outcomes equal to or more enhanced than the antibiotics regularly used during treatment or in regular periodontal treatment only [25,26]. Moreover, several studies conducted on humans have reported the palpable advantages of using NS and TQ for periodontal disease prevention and treatment besides the regular scaling on clinical, molecular, and histological levels [17,21,27,28]. As explained in the current review, such periodontal health-favoring effects of TQ and NS extracts seem to be encouraged through the distinct antibacterial, antioxidant/inflammatory, and potentially regenerative mechanisms of these rich natural compounds [27,29,30].

In recent years, lipid-based topical formulations have gained traction in drug delivery systems due to their stability, biodegradability, and non-toxic attributes, potentially enhancing the therapeutic efficacy of constituent drugs [31]. The versatility of liposomes in drug delivery offers a promising platform for carrying TQ, addressing drug solubility challenges. Liposomes may augment the therapeutic index of TQ, optimizing site-specific penetration while mitigating premature degradation and normal tissue toxicity [32]. Scaling is an important intervention in the treatment of gingivitis due to several reasons. Firstly, SC helps remove plaque and calculus deposits, which are the primary causes of gingival inflammation. By thoroughly cleaning the tooth surfaces, especially around the gumline, SC eliminates the source of inflammation and allows for gum healing [33]. Secondly, SC reduces the bacterial load in the oral cavity by removing plaque and calculus, which harbor bacteria associated with periodontal disease. This reduction in bacteria helps control the infection and inflammation in the gums, facilitating healing [34]. Thirdly, SC eliminates local irritants by smoothing the root surfaces, reducing bacterial adherence, and restoring a healthier environment for the gums [35]. At last, SC plays a crucial role in restoring periodontal health by allowing the gum tissues to heal, reattach to the tooth surfaces, and stabilize the gingival condition [36]. The purpose of this double-blinded, parallel, randomized controlled clinical trial is to assess the clinical effectiveness of topically applied 5% TQ gel. It also aims to ascertain the potential of 5% TQ gel as an adjunct to scaling (SC) in individuals diagnosed with gingivitis, compared to a placebo group.

## 2. Materials and Methods

This randomized controlled clinical trial was conducted for the evaluation to assess the clinical efficacy of TQ gel (5%) as an adjunct to SC in patients with gingivitis (n = 63). The patients were recruited from the clinic of the Faculty of Dentistry, Qassim University, and the subjects were divided into three groups. They included the following: Group I (SC + TQ gel), Group II (SC + Placebo), and Group III (Only SC-one stage prophylaxis). The random allocation of patients to these groups was determined using Stratified Randomization, ensuring an unbiased distribution of participants among the treatment arms. Allocation concealment was maintained to prevent selection bias. The patients and principal investigators were unaware of the gel preparation given to each participant, making it a double-blinded clinical trial. The gel formulations (TQ gel and placebo gel) were prepared separately, ensuring that the investigators administering the treatments and measuring the clinical outcomes were blinded to the group assignment of the patients. This approach helps minimize potential biases and ensures the integrity of the study results.

The purpose of the study was explained to the subjects; informed consent was obtained in a written format. This clinical trial was performed following the ICH (International Conference on Harmonization-E6 ‘Guideline for Good Clinical Practice’) and the Helsinki Declaration 1975, as revised in 2000. The study protocol was approved by the Institutional Review Board, Qassim University, Saudi Arabia (IRB No: EAC 104-2018). The clinical trial registration number is NCT05497895. The trial started in September 2022 and ended in November 2022. A complete oral diagnosis was performed during the screening visit by clinical assessment, and periapical radiographs, if indicated, were obtained to form a definitive diagnosis. The periodontal charting was performed and recorded in the patient’s case record.

The participants were consecutively enrolled with the following inclusion and exclusion criteria.

### 2.1. Inclusion Criteria

Patients diagnosed with gingivitis (probing <3 mm with bleeding on probing ≥10% of sites).Minimum of twenty teeth in the oral cavity.Age: 18–40 years.

### 2.2. Exclusion Criteria

Patients with systemic diseases associated with periodontal diseases, such as diabetes, cardiovascular diseases, blood dyscrasias, or diseases of the immune system, and patients requiring antibiotics before dental treatment.Patients who received antibiotic therapy in the last 3 months before the trial.Pregnant or lactating females.Patients treated with drugs such as antacids, warfarin, and cyclosporine.Presence of overhanging restorations or other contributing factors to periodontal disease.Allergy to *Nigella sativa* and/or TQ.

### 2.3. Sample Size

The sample size was recalculated using G*Power software (Version 3.0.10) at 90% power with the 0.05 alpha error based on the study conducted by Kapil et al., and it was found to be 51.29 [29]. To ensure robustness, the total sample size was increased to 63 participants, with 21 patients in each group, to compensate for possible attrition during follow-up.

### 2.4. Randomization

The selected patients were assigned to any of the three groups using the stratified randomization method.

The preparation of the lipid-based TQ gel formulation (5%) was as follows:

Thymoquinone extract (Sigma Aldrich Extract-Product No: 03416, Burlington, MA, USA) was obtained in crystalline form. TQ oral gel was prepared under complete aseptic conditions using the following components: plain oral mucoadhesive gel contained sorbitol (10% *w*/*w*), sodium para-hydroxybenzoate (0.15% *w*/*w*), hydroxypropyl methylcellulose (HPMC, 4% *w*/*w*), and distilled water. The plain gel was added to the TQ solution prepared in propylene glycol (20 mg TQ in 1 mL), adopting geometric dilution and appropriate mixing. The weight of the medicated gel was adjusted to 10 g (i.e., 0.2% *w*/*w* of TQ). The gel was adjusted to pH (7.4) by buffer, and viscosity, stability, and microbial limits were tested using appropriate analytic procedures.

### 2.5. Mode of Administration

In Group I patients, the lipid-based TQ gel (5%) was applied topically to the affected areas twice daily for two weeks following SC. The investigator performed the treatment of gingivitis based on both the patient’s compliance and the host’s response to therapy. For home care, the patients were instructed to clean and dry the affected area before the gel application. Hands were washed before and after its application. The patients were instructed not to eat for 30 min following its application as this would interfere with the drug applied to the gums. The same was carried out for Group II patients who were treated with SC followed by placebo gel application in the affected areas. Group III patients were subjected to one-stage oral prophylaxis. It was a double-blinded clinical trial as the patients were not made aware of which gel preparation was given to them, and the outcome assessors were blinded while performing clinical measurements, while the operator performed SC and gel application.

### 2.6. Treatment Compliance

A diary card was given to all the patients along with instructions on filling out the card, and a reminder to bring it on subsequent visits was mentioned. The principal investigator ensured that each patient used at least 75% and not more than 125% of the study drug throughout the study period. The drug accountability, i.e., receipt, dispensing, and return of the drug, was performed either by the principal investigator or their designee. The assessment of the primary endpoint was based on the clinical scoring conducted on weekly visits throughout the study period. Among the 63 patients enrolled in the trial, 3 patients were lost during follow-up due to non-compliance.

### 2.7. Clinical Evaluation

In this study, the bleeding index assessment was conducted as part of the clinical evaluation of the patients with gingivitis. Two clinical parameters, namely the plaque index (PI) and the papillary bleeding index (PBI), were evaluated at appropriate follow-up intervals [37,38]. The PBI was specifically used to assess the presence of bleeding upon gentle probing of the gingival papilla. The assessment was performed by two calibrated independent examiners using a periodontal probe [UNC-15 (University of North Carolina 15) probe, Hu Friedy, Chicago IL, USA]. The probe was gently inserted into the gingival sulcus around each tooth, focusing on the buccal and lingual surfaces. The bleeding index was recorded as the percentage of sites exhibiting bleeding upon probing out of the total number of sites examined. This evaluation allowed for the quantification of gingival inflammation and bleeding tendencies, providing valuable information on the efficacy of the treatment interventions and the overall periodontal health status of the patients. Adverse events, if any, with the administration of TQ gel were monitored and graded as per the intensity as mild, moderate, and severe.

### 2.8. Discontinuation of Treatment

In the case of adverse events with the use of the study drug, the patients would be withdrawn from therapy or clinical assessment. Also, when the patients suffered from significant illness or underwent surgery during the study, they were withdrawn from the trial. Non-compliant patients who did not adhere to the study protocol or for other justifiable reasons were withdrawn from the current clinical trial (1 person from each group was lost during follow-up due to lack of compliance).

### 2.9. Statistical Analysis

The data were analyzed using SPSS software (Version 20.0, IBM, Corp., Armonk, NY, USA). The normality of the data set was checked using the Shapiro–Wilk test and was found to be parametrically distributed. The mean differences in PI and PBI at two time points were compared using the paired *t*-test. The mean differences in PI and PBI scores between the groups were compared using one-way ANOVA and post hoc Tukey’s test. The statistical significance was achieved when the *p*-value was less than 0.05.

### 2.10. Inter-Examiner Reliability

The inter-examiner reliability was performed using kappa statistics, and a kappa value of 0.67 was suggestive of a substantial level of agreement between the examiners.

## 3. Results

The randomized controlled clinical trial was performed on 63 subjects diagnosed with gingivitis to assess the clinical efficacy of TQ gel (5%) as an adjunct to SC with a follow-up of 2 weeks. The PI and PBI were recorded for all patients divided into the three groups, and three subjects were lost during follow-up due to non-compliance. The average plaque scores for Group I at the baseline were (1.35 ± 0.43), Group II (1.12 ± 0.41), and Group III (1.34 ± 0.48) [Figure 1]. The paired *t*-test comparison showed a statistically significant reduction in Group I at the two-week follow-up (0.74 ± 0.30) with a *p*-value of 0.04. The placebo and SC alone groups (II and III) did not show any statistically significant reduction in the plaque scores at the two-week follow-up with *p*-values of 0.08 and 0.57, respectively [Table 1]. A one-way ANOVA test was performed for group-wise comparisons of plaque scores at baseline and 2 weeks. There were statistically significant reductions in plaque scores with Group I intervention when compared to Groups II and III with a *p*-value of 0.01 at the 2-week follow-up [Table 2]. The post hoc Tukey test for multiple comparisons of mean plaque index scores showed statistically significant differences between groups I and II (*p*-value—0.01) and groups II and III (*p*-value—0.01) at the two-week follow-up [Table 3].

The mean PBI scores at baseline for Groups I, II, and III were 0.77 ± 0.32, 0.63 ± 0.26, and 0.65 ± 0.32, respectively [Figure 2]. The paired *t*-test was performed to compare the PBI scores between the groups at two time points (baseline, 2 weeks) and the difference was statistically significant with a *p*-value of 0.05 in Group I [Table 4]. The comparison of mean PBI scores between the three groups was performed using a one-way ANOVA test and it was observed that there were significant differences in the scores at the two-week follow-up at a *p*-value of 0.05 [Table 5]. The post hoc Tukey test for multiple comparisons between the three groups at a 2-week follow-up visit showed that there were significant differences between Groups I and III (*p*-value—0.05) and between Groups II and III (*p*-value—0.05) [Table 6]. There were no significant adverse effects reported with the usage of TQ gel formulation.

## 4. Discussion

Gingivitis, a prevalent oral ailment worldwide, is often managed through chlorhexidine, a gold standard known for its anti-plaque and anti-gingivitis properties [39,40]. Nevertheless, its potential side effects, such as teeth and soft tissue staining, as well as altered taste, impede its unrestricted use [41]. Emerging from recent research, various herbal products—rosemary, paprika, Chinese ginger, curcumin, and cranberry—show promise in inhibiting plaque formation in vivo, a crucial factor in gingivitis development [42].

*Nigella sativa* contains TQ, renowned for its potent anti-inflammatory and antioxidant qualities [28]. A study by Kapil et al. [29] evaluated the clinical efficacy of 0.2% TQ gel as an adjunct to scaling and root planing (SRP) in treating chronic periodontitis. This study involved 20 subjects divided into two groups: one receiving SRP with TQ gel and the other receiving SRP alone. Significant improvements were noted in various periodontal indices, including the PI, Gingival Index (GI), Probing Pocket Depth (PPD), and Relative Attachment Level (RAL). Additionally, there was a marked reduction in subgingival microbial colonies, particularly *Porphyromonas gingivalis*, *Aggregatibacter actinomycetemcomitans*, and *Prevotella intermedia*. These findings suggest that the local application of TQ gel has strong anti-inflammatory effects and can be a beneficial adjunct in periodontal therapy. Thus, this study aimed to evaluate the clinical effectiveness of TQ gel (5%) following SC in gingivitis patients.

Our findings show a significant reduction in PI scores in Group I subjects treated with TQ gel after SRP compared to baseline. This aligns with the trends observed in the study by Kapil et al. [29], reinforcing the potential of TQ gel as an effective adjunct in periodontal care. A group-wise analysis revealed that Group I’s reduced PI scores were significantly lower than those of the placebo and SRP-only groups, with only the SRP-alone group achieving statistical significance at the two-week follow-up. In contrast, Kapil’s study did not unveil significant differences in plaque and gingival scores between the groups, which may be attributed to the different concentrations of TQ used or variations in study design.

The study by Al-Bayaty et al. [30] involving a periodontal chip containing TQ and chitosan for chronic periodontitis patients demonstrated decreased bleeding on probing scores with TQ treatment, akin to our findings. Our Group I participants experienced the greatest reduction in PBI scores among the comparison groups, with significant differences in measurements between Group I and Group III. The distinct decline in bleeding and plaque scores in Group I may be attributed to TQ’s antioxidative, anti-inflammatory, and antimicrobial properties, which collectively enhance periodontal health [43,44].

In our study, Group I exhibited the most substantial reductions in both PI and PBI scores at the two-week follow-up. Notably, a significant reduction was also observed between Group II (SRP + Placebo) and Group III (SRP-alone). The ‘Hawthorne Effect’, wherein subjects alter their behavior due to their awareness of being observed, might have influenced the outcomes in Group II. This phenomenon, supported by Feil et al. [45], could have led to improved home care adherence, thereby affecting the clinical results [46].

Our double-blinded, parallel, randomized controlled clinical trial evaluated the clinical efficacy of TQ gel as an adjunct to SRP in gingivitis patients, analyzing data from 6310 participants. While this trial uniquely assessed a novel 5% TQ gel within a liposome drug delivery system and demonstrated its efficacy over a short period, future research with larger samples and longer follow-ups could address the current study’s limitations. Notably, no adverse effects arose from TQ gel usage, further supporting its safety as a therapeutic option for gingivitis.

The current study provides compelling evidence for the efficacy of 5% TQ gel as an adjunct to SRP in the management of gingivitis. The significant reductions in PI and PBI scores observed in the TQ gel group, combined with its favorable safety profile, suggest that TQ gel may serve as a viable alternative or complementary approach to conventional therapies. Future investigations should focus on addressing the identified limitations, exploring the long-term implications of TQ gel therapy in diverse populations, and comparing its efficacy to chlorhexidine to fully elucidate its role in periodontal health.

## 5. Conclusions

The efficacy of 5% TQ gel in a novel liposome drug delivery system was clinically assessed, revealing significant reductions in PI and PBI scores in patients treated with TQ as an adjunct to SRP from the baseline to the two-week follow-up visit. There were also significant differences in PI and PBI scores favoring the TQ group when compared to the placebo and SC-alone groups. Future research with larger sample sizes is essential to validate these findings. Research can be directed toward understanding the long-term efficacy of this drug delivery system in gingivitis and periodontitis patients. Additionally, trials comparing the efficacy of TQ with the ‘gold standard’ chlorhexidine can be performed, along with identifying optimal dosage and duration for maximal reduction in plaque scores and gingival inflammation.

### 5.1. Limitations

While the study provides valuable insights, several limitations should be acknowledged:Sample Size: Although the study included 63 participants, a larger sample size may enhance the generalizability of the findings and provide more robust statistical power.Heterogeneity of Participants: Variability in participant demographics and oral hygiene practices may affect the outcomes, suggesting the need for stratified analyses in future studies.

### 5.2. Strengths

This study also presents several strengths:Innovative Drug Delivery System: The use of a novel liposome drug delivery system for TQ gel represents a significant advancement in the formulation of herbal treatments for gingivitis, potentially enhancing bioavailability and therapeutic efficacy.Comprehensive Clinical Evaluation: Multiple clinical parameters, including PI and PBI, were assessed, providing a thorough evaluation of TQ gel’s effectiveness.Double-Blinded Randomized Design: The rigorous study design minimizes bias and enhances the reliability of the results, making the findings more robust.No Reported Adverse Effects: The absence of significant adverse effects associated with TQ gel usage supports its safety as a therapeutic option for gingivitis patients.

## Figures and Tables

**Figure 1 healthcare-12-01898-f001:**
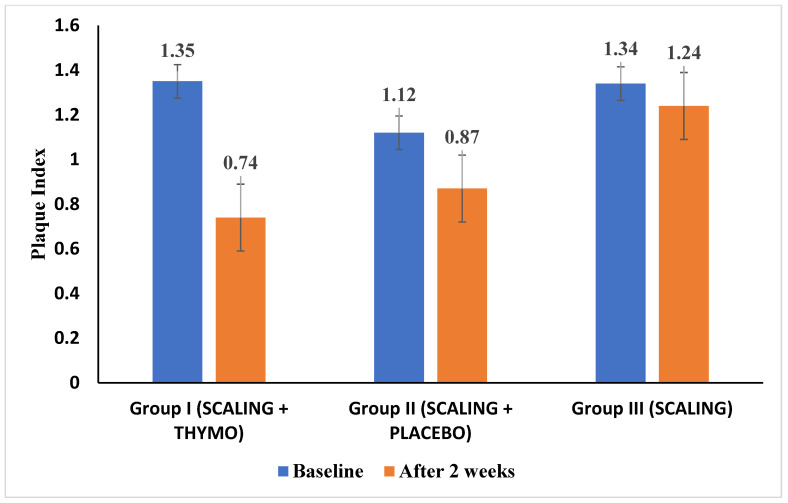
Bar diagram representing mean plaque index scores of Groups I, II, and III at two time points.

**Figure 2 healthcare-12-01898-f002:**
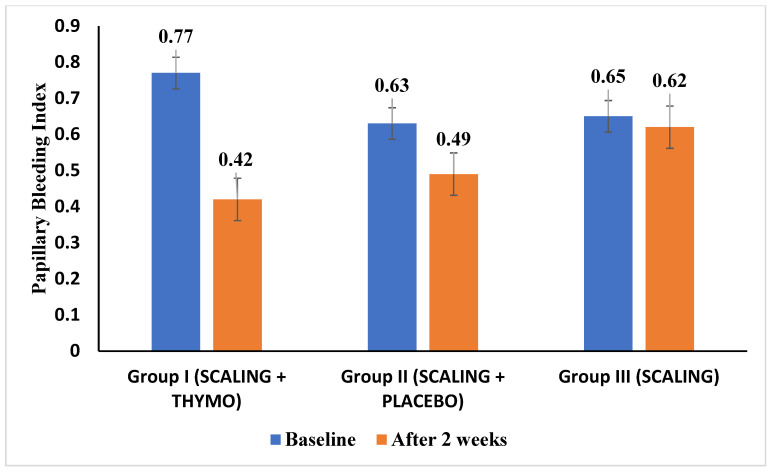
Bar diagram representing mean papillary bleeding index scores of Groups I, II, and III at two time points.

**Table 1 healthcare-12-01898-t001:** Paired *t*-test for comparison of mean plaque index scores of Groups I, II, and III at two time points.

Time Points	Groups	N	Plaque Index Score (Mean ± SD)	f Value	*p* Value
Baseline	Group I (SC + TQ)	20	1.35 ± 0.43	1.43	0.24
	Group II (SC + PLACEBO)	20	1.12 ± 0.41		
	Group III (SC ALONE)	20	1.34 ± 0.48		
At 2 weeks	Group I (SC + TQ)	20	0.74 ± 0.30	15.0	0.01 *
	Group II (SC + PLACEBO)	20	0.87 ± 0.32		
	Group III (SC ALONE)	20	1.24 ± 0.54		

* *p*-value < 0.05 considered statistically significant.

**Table 2 healthcare-12-01898-t002:** One way ANOVA test for the comparison of mean plaque index scores between Groups I, II, and III at baseline and at 2-week follow-up.

Groups	Time Points	N	Plaque Index Score (Mean ± SD)	t Value	*p* Value
Group I (SC + TQ)	Baseline	20	1.35 ± 0.43	4.57	0.04 *
	After 2 weeks	20	0.74 ± 0.30		
Group II (SC + PLACEBO)	Baseline	20	1.12 ± 0.41	1.79	0.08
	After 2 weeks	20	0.87 ± 0.32		
Group III (SC ALONE)	Baseline	20	1.34 ± 0.48	0.56	0.57
	After 2 weeks	20	1.24 ± 0.54		

* *p*-value < 0.05 considered statistically significant.

**Table 3 healthcare-12-01898-t003:** Post hoc Tukey HSD test for multiple comparison of mean plaque index scores between Groups I, II, and III at 2-week follow-up.

Time Point	Groups (I)	Groups (J)	Mean Difference (I–J)	Sig.
At two weeks	Group I (SC + TQ)	Group II (SC + PLACEBO)	−0.13	0.50
		Group III (SC ALONE)	−0.62 *	0.01 *
	Group II (SC + PLACEBO)	Group I (SC + TQ)	0.13	0.50
		Group III (SC ALONE)	−0.48 *	0.01 *
	Group III (SC ALONE)	Group I (SC + TQ)	0.62 *	0.01 *
		Group II (SC + PLACEBO)	0.48 *	0.01 *

* *p*-value < 0.05 considered statistically significant.

**Table 4 healthcare-12-01898-t004:** Paired *t*-test for comparison of mean papillary bleeding index scores of Groups I, II, and III at two time points.

Groups	Time Points	N	PBI Score (Mean ± SD)	*t* Value	*p* Value
Group I (SC + TQ)	Baseline	20	0.77 ± 0.32	6.57	0.05 *
	After 2 weeks	20	0.42 ± 0.21		
Group II (SC + PLACEBO)	Baseline	20	0.63 ± 0.26	1.28	0.26
	After 2 weeks	20	0.49 ± 0.32		
Group III (SC ALONE)	Baseline	20	0.65 ± 0.32	0.02	0.95
	After 2 weeks	20	0.62 ± 0.39		

* *p*-value < 0.05 considered statistically significant.

**Table 5 healthcare-12-01898-t005:** One way ANOVA test for the comparison of mean papillary bleeding index scores between Groups I, II, and III at baseline and at 2-week follow-up.

Time Points	Groups	N	PBI Score (Mean ± SD)	f Value	*p* Value
Baseline	Group I (SC + TQ)	20	0.77 ± 0.32	1.37	0.26
	Group II (SC + PLACEBO)	20	0.63 ± 0.26		
	Group III (SC ALONE)	20	0.65 ± 0.32		
At 2 weeks	Group I (SC + TQ)	20	0.42 ± 0.21	2.69	0.05 *
	Group II (SC + PLACEBO)	20	0.49 ± 0.32		
	Group III (SC ALONE)	20	0.62 ± 0.39		

** p*-value < 0.05 considered statistically significant.

**Table 6 healthcare-12-01898-t006:** Post hoc Tukey HSD test for multiple comparison of mean papillary bleeding index scores between Groups I, II, and III at 2-week follow-up.

Time Point	Groups (I)	Groups (J)	Mean Difference (I–J)	Sig.
At two weeks	Group I (SC + TQ)	Group II (SC + PLACEBO)	−0.07	0.50
		Group III (SC ALONE)	−0.22 *	0.05 *
	Group II (SC + PLACEBO)	Group I (SC + TQ)	0.07	0.50
		Group III (SC ALONE)	−0.15 *	0.05 *
	Group III (SC ALONE)	Group I (SC + TQ)	0.22 *	0.05 *
		Group II (SC + PLACEBO)	0.15 *	0.05 *

** p*-value < 0.05 considered statistically significant.

## Data Availability

Data are contained within the article. The original contributions presented in the study are included in the article, further inquiries can be directed to the corresponding authors.

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
