# Peer review of "An Assessment of the Clinical Efficacy of a Topical Application of 5% Thymoquinone Gel for Plaque-Induced Gingivitis Patients: A Randomized Controlled Clinical Trial"

_healthcare, 2024, doi:10.3390/healthcare12181898_

Round 1

Reviewer 1 Report

Comments and Suggestions for Authors

Dear authors,

Below I´m going to present my considerations about the manuscript:

First of all, the numbering of the bibliography, it´s better if you can order it in cardinal numerology.

In relation to introduction:

The line number 43: “Gingivitis is considered the most widespread periodontal disease that affects the emerging dental layer” , there isn´t any references. Please, add it.

In line 52 it is said: “Recent literature has unveiled the potential role of oxidative stress in the pathogenesis of periodontal disease”. It could be better if you can  add more references.

The same as line 54: “Oxidative stress prominently contributes to the transition from gingival to chronic inflammatory periodontal disease, which is often provoked by bacterial stimuli from dental plaque”. It is an affirmation that it should be supported by more references.

In line 56: “Studies have probed end products of lipid peroxidation…” only is written one reference and the subject is plural, please, add more references.

Add more references in line 63.

What is the meaning of the acronym NS that appears for first time in line 65? Please add the meaning of it. I supposed that is Nigella sativa, please confirm.

The paragraph that begins in line 64 is so long and the references only appear at the end of it, please add references in the middle.

In line 70: “Moreover, several studies conducted on humans…, please after humans, add the references of the papers.

Please change al last instead of Lastly in line 90.

In relation to material and method:

In relation to the sample calculations, it is written that the calculation of the sample was 51, but was it taken into account that the sample was going to be divided into groups?. From my point of view, only 20 patients per group it is a very low number to consider the results reliable.

In relation to clinical evaluation: Were the examinators calibrated previously? (line 183)

In relation to discussion:

In the paragraph that begins in line 268, there aren´t any references, please add it. In addition, in relation to the line “In with gastric administration of TQ [2]”, I don´t understand the number 2.

In line 275  where it is written “by Kapil et al” please add the reference after et al. The same in line 280 Al-Bayaty et al.

In general the discussion is sort, please it is important to improve this part, comparing the methodology , sample size etc with other studies.

I think is important add more limitations of the study, like the kappa value, that it is ok but noy strong, the sample size

In addition, add strengths of the trial.

Author Response

Dear Reviewers,

I would like to express my sincere gratitude for your thorough review and insightful comments on my manuscript titled “Assessment of Clinical Efficacy of Topical Application of 5% Thymoquinone Gel for Gingivitis Patients, A Randomized Controlled Clinical Trial”. Your feedback has been invaluable in refining the quality of the manuscript, and I appreciate the time and effort you dedicated to this process.

Summary of Key Findings

The study investigates the clinical efficacy of a 5% thymoquinone (TQ) gel as an adjunct to standard scaling in patients with gingivitis. Our findings indicate that the use of TQ gel significantly reduces plaque and gingival bleeding scores compared to both a placebo and scaling alone. This research contributes to the limited literature on herbal interventions in periodontal disease management and underscores the potential of TQ as a therapeutic option.

Responses to Reviewers' Comments

  1. Bibliography Numbering:

   We have revised and updated all references throughout the paper, ensuring they are listed in cardinal numerology according to the AMA reference style.

  1. Line 43 Reference Addition:

   We have added appropriate references to support the claim that “gingivitis is considered the most widespread periodontal disease that affects the emerging dental layer.”

  1. Line 52 Reference Addition:

   Additional references have been included to substantiate the statement regarding oxidative stress in the pathogenesis of periodontal disease.

  1. Line 54 Reference Addition:

   More references have been added to support the assertion that oxidative stress contributes to the transition from gingival to chronic inflammatory periodontal disease.

  1. Line 56 Reference Addition:

   We have added multiple references to support the statement regarding the end products of lipid peroxidation.

  1. Line 63 Reference Addition:

   Appropriate references have been added to the relevant section.

  1. Clarification of Acronym “NS”:

   The meaning of the acronym “NS” has been clarified as “Nigella sativa” in line 64.

  1. References in Long Paragraphs:

   We have inserted references in the middle of the long paragraph starting in line 64 for improved clarity and support.

  1. Paragraph starting in line 70 References Addition:

   References have been added after the mention of studies conducted on humans.

  1. Change from “Lastly” to “At last”:

    The wording has been changed in line 91 as per your suggestion.

  1. Sample Size Calculation:

    We acknowledge that the sample size is relatively small, which may limit the generalizability of our findings. However, we believe our results provide important preliminary insights into the efficacy of 5% TQ gel in the management of gingivitis. We have also mentioned in the paper that further research with larger sample sizes is essential to validate the findings of our clinical trial.

  1. Calibration of Examiners:

    We have included a statement confirming that the examiners were calibrated prior to conducting the clinical evaluations.

  1. References in Discussion Paragraph:

    The paragraph starting in line 296 now includes appropriate references to support the claims made.

  1. The paragraph starting in the line 295 has been renewed and properly referenced "Nigella sativa contains TQ, renowned for its potent anti-inflammatory and antioxi-dant qualities28. A study by by Kapil et al. 29 evaluated the clinical efficacy of 0.2% TQ gel as an adjunct to scaling and root planing (SRP) in treating chronic periodontitis ...."
  2. References for “by Kapil et al.” and “Al-Bayaty et al.”:

    We have added the relevant references immediately following these citations in lines 309 and 316.

  1. Improvement of Discussion Section:

    The discussion section has been expanded to provide a more comprehensive comparison of our methodology and sample size with existing studies.

  1. Strengths and Limitations:

    We have explicitly outlined the strengths and limitations of our study, including the kappa value and sample size considerations.

Summary of Changes Made

  • Revised all references to cardinal numerology in AMA style.
  • Added references throughout the introduction and discussion sections as requested.
  • Clarified acronyms and improved the structure of the materials and methods section.
  • Enhanced the discussion with more depth and proper citations.
  • Clearly outlined strengths and limitations of the study.

Thank you once again for your constructive feedback. I believe these revisions have significantly improved the manuscript, and I look forward to your further evaluation.

Reviewer 2 Report

Comments and Suggestions for Authors

Dear Authors,

Thank you for allowing to review your work. Overall, it's a good and important research work, however, have following minor comments:

1- Cite lines 102-109, 64-70

2- Lines 274-278: explain why the results contrast Kapil's (possibly the baseline scores for those in TQ intervention are higher compared to placebo and looking at post intervention graphs there is an overlapping in confidence intervals of three groups stating not much significantly different)

3- In results section please consider providing patient's socio-demographic information and all relevant information collected at the time of enrollment into the present study through summary/descriptive statistics table with each group as a column e.g. included participants clinical and socio demographic characteristics by each groups I-III. 

Comments on the Quality of English Language

Minor grammatical syntax e.g., The first time intext scaling is mentioned it is followed by SC in parenthesis. However, in later parts of text, SC and scaling are used interchangeably instead of just SC.  

Author Response

(The authors gave the same response as above.)

Round 2

Reviewer 1 Report

Comments and Suggestions for Authors

Dear Authors 

The manuscript has improved a lot.

I have no more considerations